# Intraoperative Brain Mapping in Multilingual Patients: What Do We Know and Where Are We Going?

**DOI:** 10.3390/brainsci12050560

**Published:** 2022-04-27

**Authors:** Jesús Martín-Fernández, Andreu Gabarrós, Alejandro Fernandez-Coello

**Affiliations:** 1Hospital Universitario Nuestra Señora de Candelaria (HUNSC), Neurosurgery Section, 38010 Santa Cruz de Tenerife, Spain; jesusmarfdez9@gmail.com; 2Hospital Universitari de Bellvitge (HUB), Neurosurgery Section, Campus Bellvitge, University of Barcelona—IDIBELL, 08097 L’Hospitalet de Llobregat, Spain; agabarros@bellvitgehospital.cat; 3CIBER de Bioingeniería, Biomateriales y Nanomedicina (CIBER-BBN), 08025 Barcelona, Spain

**Keywords:** multilingualism, electrical stimulation mapping, language switching, functional magnetic resonance imaging

## Abstract

In this review, we evaluate the knowledge gained so far about the neural bases of multilingual language processing obtained mainly through imaging and electrical stimulation mapping (ESM). We attempt to answer some key questions about multilingualism in the light of recent literature evidence, such as the degree of anatomical–functional integration of two or more languages in a multilingual brain, how the age of L2-acquisition affects language organization in the human brain, or how the brain controls more than one language. Finally, we highlight the future trends in multilingual language mapping.

## 1. Background from Philosophy to Intraoperative Brain Mapping

How capable is language of providing knowledge about the world if the connection between the world and language is arbitrary or conventional? This question Plato (fifth century B.C.) asked in *Cratylus*. We know that language is capable of shaping cognition, behavior, and even the form and functionality of our central nervous system [1]. However, it has always posed a challenge, not only philosophically or psycholinguistically [2,3], but also neuroscientifically. If we define multilingualism or polyglotism as the ability to speak fluently and understand two or more languages (including dialects) [4], it is estimated that more than 50% of the world’s population is multilingual [5]. Multilingual processing is among the most ubiquitous and cognitively complex tasks we perform daily [1]. Indeed, its neurobiological basis has been extensively studied in recent decades due to the rapid progress in imaging techniques, such as functional magnetic resonance imaging (fMRI) [6], tractography, and electrical stimulation mapping (ESM) [7].

Second language (L2) learning has been studied to show how it generates profound anatomical–functional neuroplastic changes in the human brain [8,9]. Among the changes induced at the cortical level, structural modulations in grey matter density have been described in the left inferior parietal lobe [10], left inferior frontal gyrus [11], temporal pole [12], and cerebellum [13]. In contrast, changes in the white matter have not been as extensively studied, although increased fractional anisotropy (FA) in multilinguals has been described by diffusion weighted magnetic resonance imaging (MRI, DT) in the superior longitudinal fasciculus (SLF), inferior longitudinal fasciculus (ILF), and uncinate fasciculus (UF) [14]. Nevertheless, the two regions that have most consistently shown multilingualism-induced changes are the corpus callosum [15,16] and the inferior occipito-frontal fasciculus (IFOF) [17,18], for which contradictory results have been described [19]. Indeed, the effects of bilingualism on the IFOF are still ambiguous. It is also important to note that these DT studies have looked for global differences at the whole-brain level by using Tract-Based Spatial Statistics (TBSS) without a TOI (tract of interest) approach. However, a whole-brain approach can lead to losing detailed participant-wise information about individual tracts. Moreover, this effect of learning on brain plasticity is not exclusive to language. There are some extraordinary phenomena observed in human nature, such as the nomadic children of the Moken tribe of the Thai coast with “supra-human” underwater vision resulting from the need to spend long periods of time diving for food [20] or the increased volume of the posterior hippocampus measured by structural MRI of London cab drivers [21], as a consequence of the need to develop an extraordinary spatial memory.

Much research on multilingualism has focused on explaining how a single cognitive system is able to manage multiple linguistic systems through mechanisms such as “language switching” [1]. However, the precise mechanisms and the specific areas that allow for the successful control of multiple languages have yet to be definitively established. Related to this, Green and Abulatebi’s [22] Adaptive Control Hypothesis posits a complex system that includes various functions such as monitoring, inhibition, task engagement, and disengagement, which are employed to varying degrees depending on the context. It has also been suggested through various fMRI studies that bilingual language control may recruit many of the same neural regions utilized for domain-general cognitive control [23]. These include the dorsolateral prefrontal cortex, which is associated with goal maintenance [24,25]; the anterior cingulate cortex and pre-supplementary motor area, linked to conflict-monitoring and attention regulation [26,27]; and the basal ganglia, which are related to functions involved in procedural memory and skill learning [22,28].

It is worth noting that ESM has been a key part of the exponential growth in the neurobiology of multilingualism in recent decades through electrical stimulation mapping (ESM) during awake surgery since it was described by Penfield et al. in 1959 [29] as a useful tool in refractory epilepsy surgery within regions involved in language processing. Nevertheless, it was Victor Horsley who first described awake craniotomy using direct cortical stimulation in 1886. The investigation of language localization with ESM during neurosurgical operations under local anesthesia has provided a different perspective on the original model of language [30] described by Paul Broca [31] and Carl Wernicke [32] based on a posterior inferior frontal Broca area and a temporoparietal Wernicke area in the 19th century. Intraoperative localization of speech is problematic in multilingual patients [33]. Various studies have shown that intraoperative brain mapping is a precise, safe, and effective technique for detecting cortical areas and white matter tracts involved in language, both in monolingual and multilingual patients [34,35,36]. In this review, we discuss some questions that remain partially unanswered, such as the degree of anatomical–functional overlap between different languages and its possible implications, how the age of L2-acquistion could affect language organization, and how we could map and preserve regions involved in language switching through ESM. Finally, we address some relevant issues concerning intraoperative brain mapping in multilinguals (Table 1).

## 2. A Key Issue: The Degree of Anatomical-Functional Integration or Separation of Two or More Languages in a Multilingual Brain

This question, which remains partially unanswered, is of utmost importance not only from a clinical and neurosurgical perspective but also from a neurobiological one. If different languages really had a complete neuroanatomical overlap, would it be necessary to monitor each of them, or would just one of them be enough? Even though the study of multilingual language processing has rapidly gained interest in recent decades, there is still an intense debate in neuropsychological–neuroimaging studies and ESM studies about this question.

From a neuropsychological point of view, a key question is how different components of language are or are not shared in multilingualism. The three most influential behavioral models of bilingual language organization all indicate that semantic systems are completely or partly shared across languages. The revised hierarchical model [37], with its focus on lexico-semantic links, and the BIA + model [38], which is focused on orthographic lexical representations, assume there is only one shared semantic system. However, this does not imply that the meaning of every word should be completely identical in every language. Indeed, the distributed feature model [39], which has been the only one that has focused in somewhat more detail on the organization of semantic representations and the factors that may influence it, assumes partially overlapping semantic features (instead of whole concepts) across languages.

Despite the relative consensus among the neural models of bilingual language processing concerning lexico-semantic organization, neuroimaging studies that investigated the hypothesis that semantic systems of both languages are represented by cortical overlapping areas have provided very divergent results, probably due to their huge methodological heterogeneity [40]. For instance, Illes et al. (1999) examined whether semantic processes in two languages are underlain by a common neural system in eight fluent English–Spanish bilingual subjects, who had learned their two languages sequentially rather than simultaneously, through a fMRI examination while participants made semantic and nonsemantic decisions about words in Spanish and English. This study focused on the cortical substrates of semantic processing. They observed that semantic activation for both languages occurred in the same cortical locations: left inferior frontal gyrus (IFG) (Brodmann’s areas (BA) 44, 45, 47), right IFG (BA 44, 45, 47), left temporal lobe (BA 22), and right middle frontal gyrus (BA 9, 46). Furthermore, no activation difference was observed in a direct comparison of semantic judgments in English and Spanish. These findings suggest that, at least to the resolution provided by fMRI, a common neural system is involved in semantic processes across both languages in the bilingual brain. Many other classical neuroimaging studies [41,42,43] used such contrast designs in which an experimental condition is compared with a control condition. Within these designs, however, the comparison between L1 and L2 may reveal the targeted cross-lingual semantic overlap, but also an overlap in peripheral untargeted processing that may result from phonology, ortography, or even mere task difficulty, because the semantic tasks are often also more difficult than the control tasks that they are compared with [44]. For this reason, to analyze the neural overlap between L1 and L2 more accurately, other approaches have been necessary, such as fMRI-adaptation and multi-voxel pattern analysis (MVPA). On the one hand, the fMRI-adaptation, through comparing the differences in the BOLD signal elicited by pairs of visual stimuli that are identical and pairs that are dissimilar, allows us to analyze clusters of neurons within the same brain region showing differential sensitivity to a feature of interest, extending the modest spatial resolution of fMRI and better enabling us to characterize how L1 and L2 is coded in the brain [45,46]. Neural overlap has been demonstrated with this approach [10]. Crinion et al. (2006) performed a study with the main goal of identifying language-dependent neuronal responses at the level of word meanings (i.e., semantics). For this purpose, they included three sets of highly proficient bilinguals: (1) 11 German–English bilinguals participated in a positron emission tomography (PET) study, while (2) 14 German–English bilinguals and (3) 10 Japanese–English bilinguals participated in fMRI-adaptation experiments. It is important to highlight that German and Japanese come from entirely separate linguistic families. Neuroimaging data showed the following results: (i) the same network of brain regions was activated for semantic decisions for both languages spoken; (ii) direct comparison of both languages did not reveal significant differences when a correction was made for multiple comparisons across the entire brain; and (iii) responses on the ventral surface of the left anterior temporal lobe mirrored the behavioral data, showing reduced activation for semantically related word pairs irrespective of whether the prime and the target word were in the same or different languages. This suggests the existence of a common semantic system in the left ventral anterior temporal lobe, which is in line with other studies about the notion that both languages are underlain by the same neuronal networks [47,48]. Conversely, other fMRI-adaptation studies have shown possible distinct neural representations [43,49]. In this regard, Chee et al. (2003) [49] reported the first study investigating the bilingual brain through fMRI-adaptation analyzing functional responses to passively viewed Chinese characters and English words in Chinese–English bilinguals. The primary aim was to determine if a word and its translational equivalent share a common neural substrate and whether words with the same meaning elicit repetition effects in areas involved in semantic processing. Meaning-sensitive effects were found in (i) left prefrontal (inferior and dorsal), (ii) left mid temporal, and (iii) left parietal regions for both languages. On the other hand, language-sensitive regions were found in (i) dorsolateral prefrontal cortex and (ii) lateral temporal areas macroscopically overlapping with meaning-sensitive areas. The higher activation observed in the mixed-language condition could be explained by (1) switching costs involving the frontal executive areas or (2) different neuronal arrays being involved in the processing of L2 and L1. The authors interpreted these results as supporting the fact that neuronal networks with differential sensitivity to semantics and language co-exist in the same broad location but are differentiable at a finer level for language.

Additionally, multivariate pattern analysis (MVPA) has been used in cross-language decoding with increasing popularity [50]. Compared to the traditional univariate method (contrast designs used by classical neuroimaging studies and fMRI-adaptation studies), which examines brain voxels in isolation, MVPA takes into account the relationships across multiple voxels and has the potential to decode fine-grained patterns of brain activity [46]. Furthermore, MVPA distinguishes patterns of neural activity associated with different stimuli or cognitive states. Van de Putte et al. (2017) [40] investigated the neural overlap between the semantic representations needed for L1 and L2 production using MVPA. For this purpose, the authors used a production task in which participants were asked to name pictures in Dutch and French. As lexical or sensory overlap was excluded across L1 and L2, the classifier could have only accurately predicted which concept was named in one language given the activation pattern for naming in the other language if semantic representations of L1 and L2 do overlap in the brain. These results provide evidence for the existence of shared semantic representations that are located in the bilateral occipito-temporal cortex and the inferior and the middle temporal gyrus. These results are in line with monolingual studies that situated L1 semantic representations along the posterior temporal regions [51,52].

In conclusion, it is important to note that multilingual neuroimaging studies have shown that different languages are represented by both shared and distinct neural patterns [46], even though most of these studies describe the sharing of the same anatomical-functional substrate of various languages in the multilingual brain.

From a neurosurgical perspective, various studies have described different results regarding this issue depending on the series of patients studied, the type of language, and the sensitivity of the neuropsychological tasks applied [33]. As far as we know, at the present time, only Bello et al. (2006) [33] and Lubrano et al. (2012) [53] have investigated polyglotism (three or more languages) through ESM including not only cortical mapping but also subcortical direct stimulation. Firstly, Bello et al. examined language cortico-subcortical organization in seven late and highly proficient multilingual patients (from three to five different languages) with a left frontal glioma. In all cases, at cortical and subcortical stimulation, three or more errors at the same site indicate that site to be essential for oral naming. They posit that sites for the first acquired language were more numerous than those for the second or further languages, in agreement with other ESM studies that have described in late bilingual patients separate areas of the cortex mediating different languages [34,54], and always distinct from other ones, which diverges from other ESM studies that have found some essential language areas shared by L1 and further languages [36]. At a subcortical level, tracts for L1 were found in four patients, with those for the other languages in three patients. Tracts for the first language were found in two sites, while those for the second or further languages were localized in just one site. The authors concluded that the location of the language sites as well as the relationship between tumor and cortical or subcortical naming site locations were not predictable and varied depending on the patient. However, it should be considered that this was a limited experiment performed on a very particular group of patients: late multilingual with a high degree of proficiency. It is possible that the late age of language acquisition in conjunction with a formal manner of learning could have contributed to the high amount of neuroanatomical divergence across languages without regions of co-localization [55]. Lubrano et al. 2012 [53] also investigated the language performance of a late and highly proficient German–French–English trilingual female suffering from a WHO grade II glioma. Direct cortico-subcortical stimulation was carried out while the patient was performing a picture-naming task in order to map and preserve her L1 (German) and French (L3), drawing the following conclusions: (i) L1 and L3 were represented in both distinct and overlapping (> 50% of all stimulation-positive sites) areas within the left (dominant) inferior frontal cortex (IFC); (ii) subcortical ESM in the white matter in the posterior and lateral depth of the surgical cavity elicited reproducible speech arrest and transient limited speech in L1 and L3, but differences between both languages were not found throughout the white matter, in contrast to Bello et al. results.

On the other hand, regarding ESM studies in bilingual patients, Lucas T et al. (2004) [36] examined a bilingual group of 22 patients with high degrees of proficiency (proficiency score > 65%) using ESM during surgical treatment of seizure disorders while an oral naming task was carried out. The most common pattern of language representation was one of discrete language-specific sites in combination with shared sites, although 40% of patients did not share sites. This evidence supports the existence of both language-specific and shared modules, which favors theories about the functional separation of languages in the multilingual brain. There were no significative differences not only in the extent of cortical representation between L1 and L2 but also in the number of essential sites for L1, L2, and shared sites. Furthermore, sites essential for naming had diverse and unpredictable anatomical locations, spread widely over the exposed cortical surface, which is consistent with findings presented by Bello et al. Lastly, the authors described that posterior regions (including temporal and parietal lobes) possess language-specific sites for L2, while anterior regions generally lack such sites. In addition, Fernández-Coello et al. (2017) [56] examined 13 multilingual patients (speaking at least three different languages) with high proficiency, including both early and late L2 multilingual. All of them were operated on because of brain tumors under ESM-awake conditions (see Figure 1 for an illustrative intraoperative image from one of the cases included in this study carried out by our research group). Results showed distinct and separate areas in 66% of all stimulated regions and overlapping areas in 34% of regions. Lastly, Połczyńska M et al. (2020) [55] conducted a systematic review in which they concluded that separate cortical specific areas dedicated to each language in both anterior and posterior language regions are the rule rather than the exception; de facto, all studies included in this review (n = 28) reported at least some areas in the brain with separate representation for L1 and L2. However, 82% of all studies also identified regions where L1 and L2 shared neural representation. In a similar vein, Połczyńska et al. concluded that two factors could explain L1 and L2 overlap: an early age of acquisition and a small linguistic distance between L1 and L2. In summary, the overwhelming and consistent findings of the separate neural organization in multilingual patients in this systematic review highlights the importance of individually mapping each language that the patient needs to communicate in post-surgically.

Bearing in mind the above review, we can conclude that (i) neuroimaging, lesion, and ESM studies coincide in that the bilingual brain cannot be considered as the sum of two monolingual systems but is a single and immensely complex brain that may differ in each case [57]; (ii) conclusions of lesion studies and intraoperative mapping differ from neuroimaging studies in several points [42]: (a) right hemisphere (non-dominant) activation is shown by fMRI studies, although language-related activation is primarily left lateralized; (b) activation in the posterior language areas is not consistently observed throughout the different neuroimaging techniques; and (iii) some neuroimaging evidence shows that semantics are shared by all the different languages that co-exist within a single brain in posterior language regions. Regarding the latter, these results agree with the theoretical linguistic perspective: there is only one language that all humans speak, and it should be defined as a biologically determined computational cognitive mechanism that assembles a potentially infinite group of hierarchically structured expressions [2]. Variations between different languages are purely at the morphological and phonological level, conversely to syntax and semantics, which remain basically invariant across all languages [58,59]. Not only have some fMRI works shown a common semantic system within the temporal lobe [10,42], but also a recent electrophysiological study [60] has demonstrated that the temporo-occipital junction and posterior superior temporal sulcus seem to be the regions involved in this type of invariant syntax-semantics across speakers. Therefore, we can hypothesize that the neural differences shown by fMRI, lesion, and ESM works between monolinguals and multilinguals reflect these extra-syntactic morphophonological transformations across different languages. However, more evidence is needed, and neuropsychological tasks during ESM or fMRI should be adjusted to study separately the different components of language.

Nevertheless, currently, it remains to be determined why imaging studies coincide only partially with lesion and ESM findings—perhaps this discrepancy in multilingualism between them could be because we study the same complex phenomenon from different points of view. On the one hand, neuroimaging studies are indirect (BOLD signal analysis) and non-invasive examinations that allow us to examine hemodynamic changes in several brain regions, either through whole-brain analysis and/or region of interest analysis (ROI analysis). Despite their non-invasiveness and high sensitivity, some limitations should be considered. For instance, some structures of the brain—more specifically, the inferior temporal gyrus and the temporal pole—may not be visualized due to interference with the magnetic field because of the air enclosed in the middle ear and the mastoid bone [58]. Furthermore, the inability to directly examine the cerebral cortex and to distinguish essential language regions from those merely participating is a crucial problem [61] that significantly reduces its precision compared to intraoperative mapping, which despite being an obviously invasive way allows us to directly measure the function of a specific area generating a transient loss of function through the use of electrical flow. Thus, it does not reach the sensitivity of neuroimaging, but it is a highly specific test that allows us to identify functionally critical language areas [62]. However, it is important to consider that intraoperative brain mapping is a surgical technique that is performed in anatomically non-intact brains affected by refractory epilepsy, cerebrovascular malformations, brain tumors, and even surgery itself, due to the ability of brain plasticity to produce anatomical changes in the location of essential language areas [61,63].

In this regard, when Dr. G. Ojemann, who was the first to map naming sites in the lateral cortex of the dominant cerebral hemisphere of bilingual patients in 1978 [57,64], was asked by a bilingual patient who was going to undergo awake surgery the next day for refractory temporal lobe epilepsy if he were going to lose some neurons in his temporal lobe, he replied: “Probably even neurons that may be active during naming and other such language tasks—but they’re not essential for them. The other stimulation sites are where naming can continue even though they are temporarily confused by the electrical buzz. The places that are merely active during naming don’t seem to be essential in the same way, even though they obviously participate in the process. We can identify them as active using microelectrode recordings, brain wave, and blood flow studies. They’re doing something that we don’t understand very well. But in epileptics where these areas cause trouble occasionally, we find that we can remove many of them without causing more trouble than we cure” (cited from Calvin and Ojemann, 1994, Chapter 14, p. 225).

In view of the above conclusions, (i) the non-dominant hemisphere has some functions that are still uncertain in multilinguism; (ii) there are divergent results mainly regarding the involvement of the posterior regions in language; and finally, (iii) bearing in mind neuroimaging and ESM studies’ constraints, it seems certain that different languages do not share all cortical and subcortical areas. Hence, although it appears clear that we only have a partial view of what is happening in our brain during multilinguistic processing, through fMRI and ESM, we can investigate what factors affect the cortico-subcortical language organization in multilinguism.

## 3. How Age of L2-Acquisition Affects Language Organization in the Human Brain

Several environmental factors have been considered to affect the neural organization of language. In fact, in multilingualism literature, there is sufficient evidence supporting the notion that the neuronal representation of language in multilingual individuals is dependent on four major factors: (1) age of second language (L2) acquisition, (2) amount of exposure and usage, (3) L2/L3/L4 degree of proficiency, and (4) linguistic distance between L1 and L2. Regarding the former, some studies [54,65] show that the cortical representation of languages is linked to the age of L2-acquisition, such that additional languages acquired at a late stage (late multilinguals are considered generally after 6–9 years old) are located in different brain areas to those of languages acquired at an earlier stage (before 6–9 years old) [56]. Regarding the latter, several studies indicate that the age limit for acquiring complete and correct pronunciation and grammar of a given language is about 6–8 years old [66,67,68], suggesting that late learners are typically less proficient than early learners, since most forms of late language acquisition are unlikely to lead to native-level competence [56].

Neuroimaging studies of bilinguals and polyglots have also highlighted the potential role of other variables such as degree of proficiency or level of exposure, showing that attained proficiency and possibly language exposure are more important than the age of acquisition as a determinant of the cerebral representation of languages in bilinguals/polyglots [57]. Functional neuroimaging with PET and fMRI offers us an indirect window into the complex mechanism of interaction of all these environmental factors [66], but various points should be taken into consideration: (i) the vast majority of neuroimaging studies have been performed on bilingual patients using fMRI [54,69,70,71]; (ii) neuroscientific studies with participants who master three or more languages presently are rare [72,73,74]; (iii) notwithstanding the fact that all fMRI studies employed the same biophysical principle of BOLD signal analysis, sometimes divergent results are obtained, probably due to the heterogeneity of the tasks used during the execution of the paradigm; (iv) fMRI is a guidance tool that has yet to be validated by ESM during awake surgery [75], and therefore ESM is not only the “gold standard” technique for localizing cortico-subcortical language areas during brain tumor or epilepsy surgery, but it is also an excellent tool that helps us to understand language organization in humans [56,61].

Perani et al. (2003) [70] designed an fMRI experiment to investigate any differences in brain activation in 11 highly proficient early bilingual subjects (Spanish–Catalan) with different acquisition ages. They concluded, through a word generation task (phonemic verbal fluency), that the age of language acquisition is a crucial factor, even in early bilinguals (at age 3) with a very high degree of proficiency for L1 and L2. De Indeed, Perani et al. observed that, in the case of word generation, more extensive cerebral activations are associated with L2 even if both L1 and L2 have been mastered with equal levels of proficiency. This could be explained by exposure, as it is less evident in bilinguals who are more extensively exposed (in percentage) to L2. Kim et al. (1997) [54] studied six early (during childhood) and six late (after puberty) bilinguals using an extended language production task (discourse), although the exact age of L2 acquisition was not provided. Kim et al. restricted an fMRI examination to Broca’s area, identifying overlapping activations for L1 and L2 in early bilinguals and spatially segregated activations in late learners. Conversely, the regions activated by L1 and L2 within Wernicke’s area overlapped in both groups of subjects, regardless of the age of L2 acquisition. However, several points bear noting to understand the differences between this experiment and the former. Firstly, no formal assessment of proficiency was made. Secondly, the exact age of L2 acquisition was not provided, meaning that these two variables could be confused in this experiment [57]. Thirdly, Kim and coworkers focused only on Broca and Wernicke’s areas. Finally, Kim et al. used an extended speech production task compared to the word generation task of Perani et al. On the other hand, Chee et al. (1999) [76] performed a word generation task–fMRI (word stem) experiment comparing functional activity among 15 early bilingual (L2 acquisition before age 6) and 9 late bilingual (L2 acquisition after age 12) Mandarin–English speakers. They did not find differences in the left inferior frontal gyrus or left middle frontal gyrus when comparing both groups, even though Mandarin not only has an ideographic writing system but there were also differences in L2’s age of acquisition. The reason for these divergent results compared to those obtained by Kim et al. could be related to (i) the different degrees of proficiency in each language (as we mentioned before when we compared Kim et al. and Perani’s et al. results) and (ii) the fact that Singapore, from where Chee’s subjects were chosen, is an integrated bilingual society in which bilingual speakers usually are highly proficient in both languages [57].

Hence, the available evidence shows that, although many factors can affect the multilingual language system, the degree of proficiency seems to be more important than age of L2-acquisition [57]. However, more evidence is needed considering that, in so many studies, the degree of proficiency and the age of L2-acquisition have not been properly assessed. In this regard, word generation and production (phonetics) in general tasks–fMRI studies—that is, in the majority of multilingual fMRI examinations—have shown that a lower level of proficiency is associated with functional activity differences in some anterior brain structures, such as Broca’s area and basal ganglia. Contrastingly, in the case of tasks focused on analyzing regions involved in comprehension, proficiency-related differences involve temporal lobes, particularly the temporal pole. In conclusion, neuroimaging studies show that when proficiency is kept constant, L2-age of acquisition per se does not seem to have a major impact on L2-brain representations [57]. Even so, functional imaging data do not bring into doubt that age of acquisition is a notable determinant of proficiency in L2.

In a different vein, from a neurosurgical standpoint, it is important to note that the majority of studies have not explicitly examined the impact of age of L2 acquisition on the degree of neuroanatomical overlap between L1 and L2 [55]. An exception is Fernández-Coello et al. (2017) [56], who focused on how age of L2-acquisition influenced the cortical spatial organization of language in 13 polyglots. The results showed that there were no significant differences in cortical extent with respect to age of language acquisition between L1 stimulation sites (47 sites), early-L2 (50 sites), and late-L2 (70 sites), which is in agreement with other ESM studies in bilingual patients [34,36] but contrary to the findings of some fMRI studies [22,70,71] discussed above. These divergences could be explained by the higher workload needed for L2 processing, which seems to be responsible for the enhanced functional activity (fMRI) in some brain areas, e.g., the dorsolateral prefrontal cortex (DLPFC) [77]. In this regard, it is important to highlight that fMRI has demonstrated limited sensitivity for identifying essential language areas (37.1–66%) [35,78]. Hence, it is perhaps not striking that highly proficient L2 speakers showed a similar fMRI representation to L1 and early L2 and further languages. Moreover, they also found that L1 and early-L2 languages tend to be largely represented within the perisylvian left hemisphere frontoparietotemporal areas and overlap. These results, in accordance with other studies [54,65,79], suggest that early-acquired languages may recruit the same neuronal networks and therefore have similar cortical language representation. Based on this, Fernández-Coello et al. hypothesized that L1 and other languages acquired early in life coinciding with the brain development process are initially fixed in classic regions (they found more language sites in Broca’s area, the posterior region of the superior temporal gyrus, and the supramarginal gyrus). Lastly, their analysis showed that age of language acquisition had a statistically significant effect on the overlapping cortical distribution: overlap was much more common among early-acquired and late-acquired languages in this set of high-proficiency subjects. Apart from that, Połczyńska et al. (2020) [55] concluded, in their systematic review, that the age of L2-acquisition seems to be a robust variable affecting the amount of overlap between L1 and L2. The second language acquired early (they used age 5 as the cutoff age) is more likely to neuroanatomically overlap with L1, while L2 learned late is more likely to be organized separately from L1, in agreement with Fernández-Coello et al. results. In this regard, Połczyńska et al. (2016) [80] demonstrated in a single case of a highly proficient polyglot (quadrilingual) with a left frontal brain tumor, who acquired L2 at an early stage (at age 5) and at a later stage her L3 and L4 (at age 15), several key findings: (1) in a highly multilingual system, there are not only shared essential language areas in classic language regions, but also specific areas for each language throughout the traditional language zones. That is, different languages may be localized in the same gross anatomical regions but in separate microanatomical systems, as they described in Broca’s area in their patient; (2) age of acquisition may differentially affect languages. The earliest acquired were the most impaired, and later learnt languages were the most resilient to damage; (3) earlier-acquired languages appeared to have a more overlapping representation, whereas the later learned languages diverged neuroanatomically. However, these results should be interpreted carefully if we consider that (i) this was a single-case report and (ii) the craniotomy exposure was not specified; thus, it is unclear if they could map all perisylvian areas.

To close, we can conclude that the acquisition of the second and further languages is a dynamic process that depends not only on the age of acquisition but also on the degree of proficiency, both being linked to each other and playing key roles in the highly variable interindividual cortical language organization in multilingual patients. This implies that ESM during awake surgery is a necessary option to map each language in order to prevent postoperative deficits [33]. Moreover, knowledge about how our brain modifies its anatomy and its functionality under the influence of various factors related to multilingualism could have important rehabilitation applications that can shape speech therapy to achieve the best results in aphasic patients.

## 4. Language Switching: A Higher-Order Cognitive Process

“I can not comunicare con you; Oggi I can not say il mio nome to you; I am a disastro today”.[81]

These sentences were what a polyglot (trilingual: Armenian L1, English L2, and Italian L3) patient answered when she was invited to describe her daily life after her stroke, with involuntary switching between languages. A CT scan showed a hypodense lesion in the white matter adjacent to the left caudate nucleus. This multilinguistic clinical case is one of the first reported involuntary switchings in literature on lesional findings [81].

The neural basis of language switching (LS) and the cognitive models of multilingualism remain controversial and partially unknown. The ability to switch in an effortless manner between languages is an intriguing and challenging topic, as well as the executive capacity to maintain that language with no interference from the rest. Conversely, when a multilingual brain becomes damaged, it could develop pathological switching, which is a phenomenon consisting of passing from one utterance or sentence to another without appropriately adapting the language in use to the given situation [56,82].

The neural evidence from functional neuroimaging research points to multiple neural regions of control that may rely upon an inhibitory mechanism of the unintended language [72] throughout a more general executive control system independently of language processing [25,49,83]. Conversely, other studies directly comparing task switching to LS suggest the involvement of language domain-specific areas in LS compared to the cognitive switching occurring when speaking only one language [10,84]. In any case, although there is still no agreement concerning the brain regions involved in LS, and a substantial amount of evidence shows divergent interpretations, recent studies support the fact that LS depends on both linguistic and non-linguistic domains. In this regard, Abutalebi and Green (2008) [72] proposed the existence of a left cortico-subcortical network (frontal–parietal–subcortical) involved in LS. This network would be formed by the following regions: (i) dorsolateral prefrontal cortex (DLPFC), which is a key mediator of cognitive control, and some fMRI studies in bilinguals have found the selective engagement of this brain region specifically to language-switching and language selection [24,25]; (ii) anterior cingular cortex (ACC), related to error detection and attention; (iii) inferior parietal lobule, involved in working memory; and iv) basal ganglia, more specifically the caudate nucleus [10], involved in language planning and lexical selection [56,85]. In this regard, the prefrontal cortex, apart from working memory resources (i.e., updating and keeping on-line the now relevant language), may work together with the ACC and basal ganglia to inhibit the interferences from the non-target language [72,86]

In addition, Kho et al. (2007) [87] described a case of pathological switching (from L1 to L2) elicited by the direct cortical stimulation of a region immediately above the pars opercularis of the left inferior frontal gyrus in a 44-year-old late bilingual (French L1, Chinese L2) man with a medium-grade of proficiency suffering from a left temporal low-grade glioma. Although in this case, this phenomenon could also be explained by the “low proficiency hypothesis” (i.e., the patient substituted L2 for L1 because of dysphasia affecting L1), the fact that the change was immediate, without pause or hesitation, and reversible argues for an executive switch mechanism between both languages. In the same vein, Moritz-Gasser and Duffau (2009) [88] reported a case of a 47-year-old, right-handed bilingual (French and English) man who underwent awake surgery for a high-grade glioma in the left posterior temporal lobe. During cortical mapping, unintentional language switching (from L1 to L2) was elicited when the posterior portion of the superior temporal sulcus was stimulated. On the other hand, during the tumor resection at a subcortical level, reproducible LS (from L1 to L2) and phonemic paraphasias were elicited when the superior longitudinal fasciculus (SLF) was stimulated. Based on these results, and bearing in mind the conclusions of fMRI and ESM studies, these same authors proposed [89] a more connectionist or hodological model of language switching organization, leaving aside the localizationism, which is understandable given the complexity of this phenomenon. In this model, LS is underlain by (i) an executive control system formed by SMA-ACC connected to DLPFC, and from here, there is reciprocal information transmission with basal ganglia, especially with the left caudate nucleus along the cortico-striatal-thalamo-cortical (CSTC) pathway. This system might control a (ii) specific language subcircuit that involves three main epicenters: (1) the left inferior frontal gyrus (the classic Broca’s area), (2) supramarginal gyrus–angular gyrus (inferior parietal lobe); (3) posterior superior temporal areas–fusiform gyrus, which would be interconnected by the superior longitudinal fasciculus. In addition, Lubrano et al. (2012) [53] reported further evidence of the role of the left prefrontal cortex (specifically the posterior and lateral part of the middle frontal gyrus) in eliciting involuntary language switching (from L3 to L2) while stimulating this brain region in a 31-year-old trilingual (German L1, English L2 and French L3) female with a left prefrontal grade II glioma. In addition, Sierpowska et al. (2013) [90] reported two cases of bilingual patients undergoing awake craniotomy using ESM. In the first one, the resection of the left lateral middle frontal gyrus (MFG) elicited involuntary switching (from L1, Spanish to L2, Catalan). This area, because of the grade and localization of the tumor, had to be removed, resulting in post-surgical pathological switching. The second one, whose critical LS-related areas could be mapped and preserved, showed an altered performance on the LS-naming task when direct cortical stimulation was applied to the left caudal MFG. These results confirm the role of MFG in LS. They interpreted these findings considering the proposed role of the MFG in mediating cognitive control in bilinguals throughout the interaction between a top-down selection-suppression mechanism and a local inhibitory mechanism in charge of changing the degree of selection–suppression between different lexicons [91]. From the same group, Sierpowska et al. (2018) [92] conducted an LS-ESM paradigm assessment in nine Spanish–Catalan bilingual patients undergoing awake brain tumor surgery that allowed a systematic evaluation of externally triggered LS synchronously with direct cortical stimulation (see Figure 2a for a schematic picture and Figure 2b for an intraoperative image of one of the patients included). ESM showed that LS-related areas were mainly distributed across the left MFG (10 sites within MFG versus 6 sites within IFG), specifically on its posterior region (i.e., the posterior part of BA 9 and the posterior and inferior part of BA 8—anterior premotor cortex), as opposed to language naming sites, which were placed principally within the left IFG. In six patients, a larger craniotomy was performed, exposing the superior part of the left temporal lobe or the inferior part of the left parietal lobe. This enabled the authors to identify LS sites in the left superior temporal gyrus (STG) and the supramarginal gyrus (SMG). On the other hand, the fMRI experiment showed significant clusters of activation in the left IFG (mainly pars triangularis) and left MFG (in its posterior region). In addition, activations within the left superior frontal gyrus (SFG) were found to be relevant for LS in five of eight cases and in the ACC in four of eight patients. It is important to highlight that fMRI and ESM results did not always overlap. This systematic effect of the left posterior MFG across different functional brain mapping modalities (fMRI and ESM) provides evidence that MFG might be a key mediator for cognitive control in bilinguals, supporting the notion that LS is a very demanding task that shares features with other kinds of cognitive control conditions [72,91,93]. It is important to bear in mind that the role of basal ganglia or the white matter tracts underlying MFG and IFG in LS were not examined through ESM in this study. Therefore, further studies using subcortical stimulation in white matter will be important to understand their possible role in LS.

To summarize, although the exact neural mechanisms underlying language switching remain unknown, the evidence so far shows that the different languages present in a multilingual brain seem to be co-activated in parallel, at the same time as certain brain regions involved in executive functions that would inhibit the interference of unintended language. Apart from that, ESM during awake surgery has been shown to be a safe, feasible, and effective procedure for mapping the brain regions responsible for LS to prevent the possible post-surgical appearance of pathological LS. However, more fMRI and ESM studies are needed to understand the cortico-subcortical language switching organization in the multilingual brain.

## 5. Awake Intraoperative Mapping: A Look towards the Future

To cope with challenges resulting from migration and globalization, multilingualism has become an important skill [94] since it promotes education, cognitive health [95,96], and sociocultural and economic inclusion [97]. The fact that multilingual people are the rule rather than the exception has important neurosurgical repercussions, given how problematic language localization is in those people who are fluent in different languages [33]. Nevertheless, the monolingual brain and the monolingual language processing model are still considered as the norm, even accepting, more often than not, the present obsolete classic model of language, not only in neurocognitive models but also in clinical practice. This could be partly explained by the mixed and inconsistent findings and, to a certain extent, controversies regarding the functioning and neural underpinnings of language processing [94], as well as the significant divergence between the results shown by lesional studies, neuroimaging investigations, and awake intraoperative mapping studies. This divergence does not yet have a straightforward explanation.

Given that many neurosurgical diseases can affect brain regions involved in language, a major goal of neurosurgery is to map and preserve all of a patient’s languages [57,61], also considering that awake intraoperative mapping has been shown to be safe, precise, and reliable method of detection of functional cortical areas and white matter pathways [30,34,35,36,98]. It has been used during surgery for gliomas to detect cortical and subcortical sites associated with language to minimize the morbidity and increase the quality of resection [33,99,100]. Awake craniotomy is a “gold standard” for neurosurgical interventions requiring tissue resection close to “eloquent areas” of the brain [101,102], and the European Federation of Neurological Societies–European Association for Neuro-Oncology (EFNS-EANO) guidelines for low-grade glioma issued in 2010 in Europe identified brain mapping during awake craniotomy as a method to obtain Class II evidence [103].

### 5.1. The Conceptual Evolution of Broca’s and Wernicke’s Areas Is Needed to Understand Multilingualism 

Despite significant advances in recent decades in the neuroscience of language, and specifically in multilingualism, the classical model of language is still used in clinical daily practice. Since the late 19th century, when Broca described the classic description of two patients with expressive aphasia after injury to the inferior frontal gyrus [31], two major language-related cortical regions have been identified: Broca and Wernicke areas interconnected by the arcuate fascicle. The anatomical and functional definition has evolved significantly over time up until the current day with the language dual-stream model. It is essential, even more from a multilingual viewpoint, to know this conceptual evolution because of the neural complexity underlying multilingualism.

Anatomically, the Broca–Wernicke model is incomplete and no longer adequate to understand the neural basis of multilingualism, as it does not represent the distributed connectivity relevant for language. Functionally, in agreement with some studies on motor speech output [104] and awake craniotomy with intraoperative mapping for low-grade glioma studies [61], findings support the fact that the so-called Broca’s area is not the speech output region, and it would also seem that it is a supramodal hierarchical processor, challenging the prevailing notion of Broca’s area as a motor area, per se [61,105]. Interestingly, Tate et al. (2014) [61] reported that crucial epicenters for speech output were not lateralized to the dominant hemispheres, although it is important to highlight that (i) subjects included in the study were not bilingual or multilingual; (ii) all the cases were brain tumor patients, which may involve an anatomical–functional distortion, not only because of space-occupying lesions but also because it has been demonstrated that low-grade gliomas and surgery, by itself, induce brain plasticity [63,106]. Despite this, these results could have crucial implications for brain surgery, and this conclusion has the potential to change neurosurgeons’ traditional surgical philosophy, even more in multilinguism intraoperative brain mapping. Nonetheless, more evidence is needed in this challenging field of neuroscience. From a functional point of view, although traditionally posterior brain regions were held to support language comprehension, modern imaging and neuropsychological studies coincide on the conclusion that this region plays a much larger role in speech production, more specifically in phonologic retrieval, and is not critical for word comprehension [44]. In this respect, other cortical areas, including the so-called Broca area and the dorsal premotor cortex, are also important for the process of language comprehension [107].

### 5.2. Subcortical Pathways in Multilingualism: A Matter of Vital and Yet Partially Unknown Importance

Since the arcuate fasciculus was first reported in 1822 by Reil and Burdach [108], anatomical, neuroimaging (fMRI and tractography), and ESM studies have contributed extensively to the knowledge of white matter and its involvement in language. Despite this, subcortical fibers remain a significantly unknown challenge. To the best of our knowledge, presently, only two ESM studies [33,53] have described subcortical fiber’s involvement in multilingualism. On the one hand, Bello et al. (2006) [33] reported seven cases of late and high proficient multilingual patients (from three to five different languages) with a left frontal glioma. After direct cortical stimulation, subcortical mapping was conducted while oral naming tasks were performed testing each language sequentially, with the following results: (i) two tracts were found for L1 in three patients and one in one patient; (ii) only one tract for the other languages (L2+) was located in three patients; and (iii) during resection, no functional fibers were found posteriorly and laterally. However, it is important to highlight that these results come from a seven-case experience in a very specific group of multilinguals: late polyglots with high proficiency levels. On the other hand, Lubrano et al. (2012) [53] reported a single case of a trilingual patient suffering from a low-grade (II) glioma who underwent an awake craniotomy with cortico-subcortical mapping of L1 and L2 while a picture naming task was being performed. Subcortical ESM in the white matter in the posterior and lateral depth of the surgical cavity elicited reproducible language disturbances (speech arrest and transient limited speech in L1 and L3), but no difference between L1 and L3 processing was found in the subcortical structures.

Apart from that, regarding the phenomenon of plasticity, it should be considered that some studies have shown that the plasticity of cortices is far stronger than that of subcortical fibers [61,109], which could have great repercussions in neurosurgery. However, subcortical fibers also have plasticity; in fact, different fiber locations also have different plasticities. Herbet et al. (2016) [110] found that the anterior superior longitudinal fasciculus/arcuate fasciculus (SLF/AF) had stronger plasticity than the other parts of the SLF/AF. Therefore, regarding these fibers with low plasticity, subcortical mapping should be done to protect white matter.

In conclusion, although subcortical mapping has some limitations,—(i) the results could be affected by the angle of stimulation and density, (ii) fatigue as a result of time increment in language mapping [111,112], and (iii) lack of evidence of the involvement in language processes through fMRI, tractography, and ESM studies—subcortical direct stimulation is a key tool in awake surgery. In fact, post-operative language deficits could not be completely avoided by only performing a cortical mapping.

### 5.3. Fatigue and Time Limitation: A Multilinguism-Brain Mapping Challenge

With an increasing number of patients who are multilingual, the adequate protection of each language function has become an important issue in neurosurgery, since the major challenge of intraoperative mapping in an awake patient is the time limitation: patients usually become tired after 2 h of testing [62,113]. The conventional language mapping protocol in multilingualism recommends mapping each language serially [33,114], which means a proportional increase in the time during which the patient must be awake while the different neuropsychological tests are performed, and the cooperation of patients worsens with increasing time [114] because of fatigue or tiredness, which is a reason for the decline in DCS accuracy [103]. To avoid this, Weng et al. (2021) [114] proposed a new protocol for language mapping in Chinese patients based on the knowledge that, compared to adults in Western countries [56], it is rare for Chinese adults to acquire a second language in the early stage (before 7 years old). This means, based on the aforementioned fMRI and ESM studies’ results (see Section 3), that those who learned a second language after 7 years old showed a wider range of activated cortices in non-native language tasks [36,56,70,115,116]. Moreover, there is evidence that there is an optimization process in the language network during the learning process consisting of a gradual shrinking of activated cortex formation, resulting in an optimized and therefore more efficient network [117,118]. This finding suggests that early exposure and acquisition of a language (L2, L3, L4…) requires a small range of cortices, while more extensive cortices are required for the recruitment of recently acquired languages or those that the person is unskilled at [119]. Based on this, the authors propose that, regarding Chinese patients who acquired more than three languages (especially any non-native languages acquired in the late stage), intra-operative mapping of only the cortical distribution of the native language and the least proficient acquired language is sufficient to protect all languages (native and non-natives). As far as we know, there are no other studies apart from this one that have proposed possible solutions to the increasing time in intraoperative mapping in multilingualism; however, the importance of intraoperative mapping of the subcortical tracts should be taken into account, given that their injury can cause language impairments [33,100] despite adequate cortical mapping. In conclusion, simplifying the protocol of cortico-subcortical language mapping in multilingualism is a key point to explore in order to protect each language function.

### 5.4. Topographical Anatomy and/or fMRI-Guided Surgery alone Cannot Predict the Language-Related Areas

On the one hand, multilingual ESM studies [33,36,64,120] have demonstrated that essential language sites are focally discrete, both in monolingual and multilingual patients. In fact, many studies have described the close proximity (within less than 1 cm) between these areas and silent sites where no naming errors occur [36]. As a matter of fact, presently, it remains unclear whether conventional neuroimaging techniques have the capability to demonstrate such separations when language essential areas are in close proximity [49]. Evidence shows that the relationship between ESM language sites and fMRI activation centers is a matter of considerable debate [35,121]. It is important to highlight that not all fMRI paradigms have been able to resolve spatial separations between multilingual language processing areas [76,122]. Moreover, there are significant methodological problems in comparing functional MRI activation and intraoperative mapping; i.e., the task paradigm necessary for good activation is the repetitive performance of a task, whereas the effect of stimulating the cerebral cortex is a single event [123]. Yetkin et al. (1997) showed in a series of 28 non-multilingual patients that the area of activation on fMRI during language-tasks and the site mapped intraoperatively during the performance of similar tasks demonstrated spatial discrepancies of up to 2 cm. The spatial extent of fMRI imaging activation is partly related to the statistical threshold at which the activations are plotted. Lower statistical thresholds typically reveal activations with spatially larger extensions (and more false-positive results) [124]. This could help explain why these studies [124,125] reveal the occurrence of both false-positive areas (areas where fMRI activation is present, but no ESM site is found) and false-negative areas (areas where fMRI activation is absent, but ESM language sites are found). Therefore, fMRI can be used to visualize language processing areas in the cortex, but those areas are not necessarily essential for it, unlike intraoperative mapping, which allows us to locate the essential areas for language with significant specificity [36,61]. Apart from this, ESM studies show that language sites in multilingual patients are widely distributed across the cortical surface.

The evidence shows that fMRI has become a useful clinical tool for the surgical management of intraoperative brain mapping cases [126]. fMRI can identify high-risk eloquent areas located proximal to a tumor, which is very helpful for presurgical planning, including the safest entry point and the most adequate approach, even assisting surgeons in maximizing the benefit-to-risk ratio of the surgery [127,128]. Nowadays, as a general rule, fMRI activity maps are visualized on a separate display during the operation in order to assist the neurosurgeon when intraoperative brain mapping is being performed. In bilingual patients, although fMRI language mapping allows for the consideration of potential functional areas that may be neglected with L1-only fMRI mapping [129], its accuracy and utility in neurosurgical planning remains partially unknown because of the variability in statistical thresholding, which influences the extent of spatial activation and affecting quantitative analysis of fMRI data and the limited spatial resolution per se. Furthermore, sometimes the usefulness of this tool can decrease because of a phenomenon called “brain shift”, which occurs during the craniotomy, resulting in spatial gaps between the preoperatively obtained fMRI map and nerve fibers displayed on the navigation system during surgery [103]. Brain shift in cases of space-occupying lesions such as glioma may be as significant as 1–2 cm, although it can be as small as 50 mm [130]. The amount of this brain shift depends on cerebrospinal fluid drainage, craniotomy size, degree of head rotation, use of drugs, errors in tracking the surgical tools, tissue loss from tumor resection, et al. [131,132,133]. Although numerous studies, including MRI based approaches, computed tomography (CT), and ultrasound with image registration techniques, have described methods to try to correct brain shift, there is currently no tool that allows the co-visualization of the fMRI brain map and DCS data intraoperatively, which as we have previously described is the “gold standard” in the neurosurgical management of tumors close to eloquent areas. All in all, without brain shift compensation, neuronavigation systems cannot be trusted at critical steps of the surgical procedure, e.g., the identification of the deep tumor margin [130]. Not only because of this but also because of the complexity that multilingualism entails for neuroscience, both from the point of view of neuroimaging and from neurosurgery, it represents an even greater challenge in multilingual patients, although to the best of our knowledge, at this point in time, there are no studies reporting “brain shift” data in neurosurgical cases of multilingualism.

We conclude that anatomical topography or functional imaging alone cannot predict the location or the absence of essential language sites in multilingual patients. Hence, direct stimulation mapping remains the best method to identify those areas that are essential naming sites for one or more languages [36].

## 6. Conclusions

Neurobiological evidence shows that, in multilingualism, there are shared cortico-subcortical modules between several languages as well as language-specific modules. Nevertheless, significant variability among subjects and the lack of high specificity of pre-surgical fMRI mapping and tractography examinations means that intraoperative cortico-subcortical mapping is an effective and potentially necessary tool to avoid possible postoperative language impairment in multilingual patients. However, it is always of vital importance to individualize each patient on a case-by-case basis.

## Figures and Tables

**Figure 1 brainsci-12-00560-f001:**
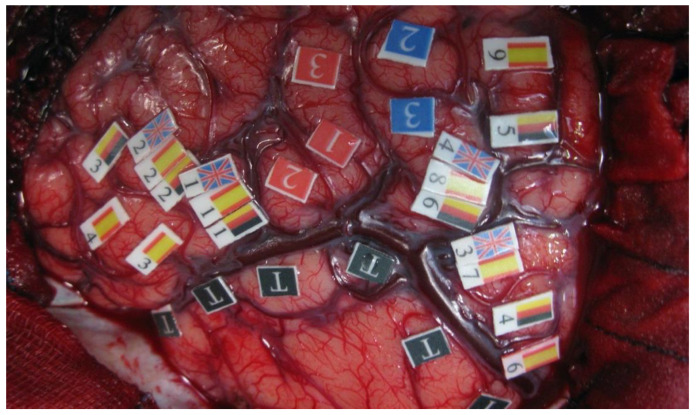
An intraoperative view of the window exposed by a left frontoparietotemporal craniotomy. Red numbers (1–2–3) represent the motor stem on the precentral gyrus. Blue numbers (2–3) represent the sensory stem on the postcentral gyrus. The areas marked with flags represent those regions in which language disturbances were elicited during the direct cortical stimulation, either specific to one language or shared among several of them. We can see at the ventral premotor cortex that it is being shared by English, German, and Spanish.

**Figure 2 brainsci-12-00560-f002:**
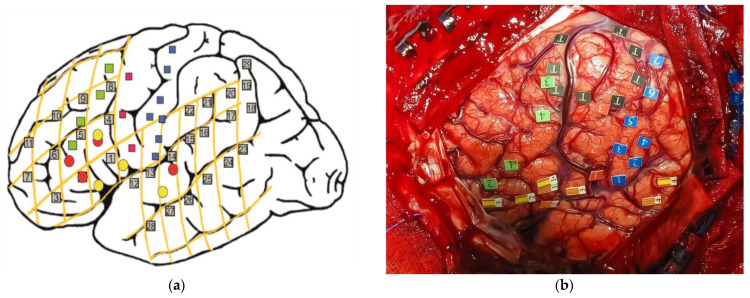
(**a**) Schematic picture representing where involuntary language switching (green), motor response (red), and sensory response (blue) were elicited by direct cortical stimulation. Essential areas for both languages are represented by red (Spanish) and yellow colors (Catalan). (**b**) A photograph was taken during surgery after the completion of ESM in the same patient. The area enclosed by flags with the letter T represents where the tumor was located. Image from the LS-patient series from Sierpowska and Fernandez-Coello (2018) [92].

**Table 1 brainsci-12-00560-t001:** Key questions about mapping in multilingual patients.

The degree of anatomical–functional integration or separation of languages
How the age of L2-acquisition does affect language organization
Neural underpinnings underlying language switching in multilinguals
Cortical and subcortical pathways in multilingualism: beyond Broca and Wernicke
Multilingual brain mapping technique: a gold standard with limitations

## Data Availability

All the data reported is in the references below.

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
