# Peer review of "Intraoperative Brain Mapping in Multilingual Patients: What Do We Know and Where Are We Going?"

_brainsci, 2022, doi:10.3390/brainsci12050560_

Round 1

Reviewer 1 Report

I suggest making a table for each main question.

Please make a concise section of conclusion.

I suggest asking an English native speaker to edit the language. Some sentence structures are weird. This is an example, (not the only weird sentence:  lines 376-7 However, we should carefully interpret these results
because in so many studies it has not been correctly assessed either proficiency or age of
L2-acquisition.

lines 417-424: The role of l2 acquisition and language proficiency is not clear, the paragraph is confusing, please reformulate

on this, Fernández-Coello et al.’s hypothesized that L1 and other languages acquired 417
early in life, are initially fixed in classic regions (they founded in Broca’s area, posterior
418
superior temporal gyrus and the supramarginal gyrus). Lastly, the analysis showed that
419
age of language acquisition had a statistically significant effect on the overlapping corti-
420
cal distribution: overlap was much more common among early-acquired and
421
late-acquired languages, what supports the hypothesis that proficiency, more than the
422
age of acquisition, seems to be the predictor of early- and late-acquired language-related
423
sites, bearing in mind that their group of patients was a set of highly proficient group.
424

Please verify the text for some minor spelling issues and typos such as an extra e at the end of important.

Author Response

Response to Reviewer 1 Comments

Point 1: I suggest making a table for each main question.

Response 1: A table has been included with all the main key questions discussed in the review

Point 2: Please make a concise section of conclusion.

Response 2: A conclusion section has been introduced

Point 3: I suggest asking an English native speaker to edit the language. Some sentence structures are weird

Response 3: We asked a native English colleague for revising our use of English along the paper that we submitted to the journal.

Point 4: lines 417-424: The role of l2 acquisition and language proficiency is not clear, the paragraph is confusing, please reformulate

Response 4: We reformulated the unclear role of L2-age of acquisition, concretely in 417-424 lines as suggested

Point 5: Please verify the text for some minor spelling issues and typos such as an extra e at the end of important

Response 5: We checked the whole manuscript to rule out all the spelling issues and typos

Reviewer 2 Report

The authors catalogue current insights into multilingualism in the brain. This is indeed an under-studied topic, as the authors note, in particular given that the majority of humans on the planet speak multiple languages. Please address some minor conceptual issues.

An architectural point: The authors consult various models that discuss the psycholinguistic basis of multilingualism. However, they should discuss a more fundamental point from generative linguistics: That there is effectively only one language, that all humans speak, and that variations between different languages are purely at the morphophonological level – core operations at grammar/syntax and semantics are basically invariant across all languages. As such, the authors need to stress that neural differences (white matter, etc) between monolinguals and multilinguals reflects these extra-syntactic morphophonological transformations. Interestingly, the fact that bilingual usage mirrors control network activity in domain-general systems supports this view (as the authors discuss). Indeed, the evidence cited on lines 116-127 suggests that there is only one underlying semantic system between English-Spanish speakers for their languages, too. In this connection, the authors could cite recent intracranial work showing that posterior superior temporal sulcus seems to be the region involved in this type of invariant syntax-semantics across speakers, which seems to accord with some of the work they discuss elsewhere in the manuscript: https://www.jneurosci.org/content/early/2022/02/23/JNEUROSCI.1575-21.2022.

The authors could also cite more general reviews of intraoperative language mapping, which set an important architectural foundation for L1-L2 correspondences: https://pubmed.ncbi.nlm.nih.gov/24300983/.

Minor: line 74 ‘However, was Victor Horsley’ >> However, it was Victor Horsley.

Line 261 ‘we can objectify that’ >> we can conclude that

Line 453: ‘a high cognitive order process’ >> a higher-order cognitive process

Author Response

Response to Reviewer 2 Comments

Thank You very much for giving us the opportunity to review our manuscript

Point 1: An architectural point: The authors consult various models that discuss the psycholinguistic basis of multilingualism. However, they should discuss a more fundamental point from generative linguistics: That there is effectively only one language, that all humans speak, and that variations between different languages are purely at the morphophonological level – core operations at grammar/syntax and semantics are basically invariant across all languages. As such, the authors need to stress that neural differences (white matter, etc) between monolinguals and multilinguals reflects these extra-syntactic morphophonological transformations. Interestingly, the fact that bilingual usage mirrors control network activity in domain-general systems supports this view (as the authors discuss). Indeed, the evidence cited on lines 116-127 suggests that there is only one underlying semantic system between English-Spanish speakers for their languages, too. In this connection, the authors could cite recent intracranial work showing that posterior superior temporal sulcus seems to be the region involved in this type of invariant syntax-semantics across speakers, which seems to accord with some of the work they discuss elsewhere in the manuscript: https://www.jneurosci.org/content/early/2022/02/23/JNEUROSCI.1575-21.2022.

Response 1: Thank you for this comment. We discuss in point 2, page 6: “iii) some neuroimaging evidence shows that semantics are shared by all the different languages that co-exist within a single brain in posterior language regions. Regarding the latter, these results agree with the theoretical linguistic perspective: there is only one language, that all humans speak, and it should be defined as a biologically determined computational cognitive mechanism that assembles a potentially infinite group of hierarchically structured expressions [2]. Variations between different languages are purely at the morphological and phonological level, conversely to syntax and semantics, which remain basically invariant across all languages [58,59]. Not only have some fMRI works shown a common semantic system within the temporal lobe [10,42] but also a recent electrophysiological study [60] has demonstrated that the temporo-occipital junction and posterior superior temporal sulcus seem to be the regions involved in this type of invariant syntax-semantics across speakers. Therefore, we can hypothesize that the neural differences showed by fMRI, lesion and ESM works between monolinguals and multilinguals reflect these extra-syntactic morphophonological transformations across different languages. However, more evidence is needed and neuropsychological tasks during ESM or fMRI should be adjusted to study separately the different components of language.”

Please note that reference number 60 corresponds to https://www.jneurosci.org/content/early/2022/02/23/JNEUROSCI.1575-21.2022

Point 2: The authors could also cite more general reviews of intraoperative language mapping, which set an important architectural foundation for L1-L2 correspondences: https://pubmed.ncbi.nlm.nih.gov/24300983/.

Response 2: We have included as general review of intraoperative language mapping, the reference number 62 corresponding to https://pubmed.ncbi.nlm.nih.gov/24300983/.

Point 3: Minor: line 74 ‘However, was Victor Horsley’ >> However, it was Victor Horsley.

Response 3: Minor correction: “…it was Victor Horsley”

Point 4: we can objectify that’ >> we can conclude that

Response 4: Minor correction: “we can conclude that”

Point 5: ‘a high cognitive order process’ >> a higher-order cognitive process

Response 5: Minor correction: “a higher-order cognitive process”

Thank you for these important comments.